# Tight Analysis of Extra-gradient and Optimistic Gradient Methods For Nonconvex Minimax Problems

**Pouria Mahdavinia**
Pennsylvania State University
pxm5426@psu.edu

**Yuyang Deng**
Pennsylvania State University
yzd82@psu.edu

**Haochuan Li**
Massachusetts Institute of Technology
haochuan@mit.edu

**Mehrdad Mahdavi**
Pennsylvania State University
mzm616@psu.edu

## Abstract

Despite the established convergence theory of Optimistic Gradient Descent Ascent (OGDA) and Extragradient (EG) methods for the convex-concave minimax problems, little is known about the theoretical guarantees of these methods in nonconvex settings. To bridge this gap, for the first time, this paper establishes the convergence of OGDA and EG methods under the nonconvex-strongly-concave (NC-SC) and nonconvex-concave (NC-C) settings by providing a unified analysis through the lens of single-call extra-gradient methods. We further establish lower bounds on the convergence of GDA/OGDA/EG, shedding light on the tightness of our analysis. We also conduct experiments supporting our theoretical results. We believe our results will advance the theoretical understanding of OGDA and EG methods for solving complicated nonconvex minimax real-world problems, e.g., Generative Adversarial Networks (GANs) or robust neural networks training.

## 1 Introduction

In this paper, we consider the following minimax problem:

$$\min_{\boldsymbol{x} \in \mathbb{R}^d} \max_{\boldsymbol{y} \in \mathcal{Y}} f(\boldsymbol{x}, \boldsymbol{y}) \tag{1}$$

where $\mathcal{Y}$ could be a bounded convex or unbounded set, and the function $f : \mathbb{R}^d \times \mathcal{Y} \to \mathbb{R}$ is smooth and strongly-concave/concave with respect to $\boldsymbol{y}$, but possibly nonconvex in $\boldsymbol{x}$. Minimax optimization (Problem 1) has been explored in a variety of fields, including classical game theory, online learning, and control theory [2, 50, 21]. Minimax has emerged as a key optimization framework for machine learning applications such as generative adversarial networks (GANs) [14], robust and adversarial machine learning [46, 37, 15], and reinforcement learning [54, 43].

Gradient descent ascent (GDA) is a well-known algorithm for solving minimax problems, and it is widely used to optimize generative adversarial networks. GDA performs a gradient descent step on the primal variable $\boldsymbol{x}$ and a gradient ascent step on the dual variable $\boldsymbol{y}$ simultaneously in each iteration. GDA with equal step sizes for both variables converges linearly to Nash equilibrium under the strongly-convex strongly-concave (SC-SC) assumption [28, 12], but diverges even under the convex-concave (C-C) setting for functions such as bilinear [22, 38].

Given the high nonconvexity of practical applications such as GANs, exploring convergence guarantees of minimax optimization algorithms beyond the convex-concave (C-C) setting is one of the canonical research directions in minimax optimization. Several algorithms with convergence guarantees beyond the C-C domain have been explored in the literature. Alternating Gradient Descent Ascent

36th Conference on Neural Information Processing Systems (NeurIPS 2022).

| Algorithm | NC-C | | NC-SC | |
|---|---|---|---|---|
| | Deterministic | Stochastic | Deterministic | Stochastic |
| PG-SVRG [44] | - | $\tilde{O}(\epsilon^{-6})$ | - | - |
| HiBSA [36] | $O(\epsilon^{-8})$ | - | - | - |
| Prox-DIAG [48] | $\tilde{O}(\epsilon^{-3})$ | - | - | - |
| Minimax-PPA [31] | $O(\epsilon^{-4})$ | - | $O(\frac{\sqrt{\kappa}}{\epsilon^2})$ | - |
| ALSET [4] | - | - | $O(\frac{\kappa^3}{\epsilon^2})$ | $O(\frac{\kappa^3}{\epsilon^4})$ |
| Smoothed-AGDA [52] | - | - | $O(\frac{\kappa}{\epsilon^2})$ | $O(\frac{\kappa^2}{\epsilon^4})$ |
| GDA [30] | $O(\epsilon^{-6})$ | $O(\epsilon^{-8})$ | $O(\frac{\kappa^2}{\epsilon^2})$ | $O(\frac{\kappa^3}{\epsilon^4})$ |
| OGDA/EG (Theorems 4.2, 4.4, 4.8, 4.9) | $O(\epsilon^{-6})$ | $O(\epsilon^{-8})$ | $O(\frac{\kappa^2}{\epsilon^2})$ | $O(\frac{\kappa^3}{\epsilon^4})$ |

Table 1: A summary of prior and our convergence rates in nonconvex-concave (NC-C) and nonconvex-strongly-concave (NC-SC) minimax optimization. For NC-C, we assume $f(\boldsymbol{x}, \boldsymbol{y})$ is $\ell$-smooth, $G$-Lipschitz in $\boldsymbol{x}$, and concave in $\boldsymbol{y}$, and for NC-SC we assume $\ell$-smoothness, and $\mu$-strong concavity in $\boldsymbol{y}$, where $\kappa = \ell/\mu$ denote the condition number.

(AGDA) is one of these methods demonstrated to have excellent convergence properties beyond the C-C setting [51, 52, 6]. Additionally, two alternative powerful algorithms are Extragradient (EG) and Optimistic GDA (OGDA), which have recently acquired prominence due to their superior empirical performance in optimizing GANs compared to other minimax optimization algorithms [28, 8, 38]. Spurred by the empirical success of EG and OGDA methods, there has been a tremendous amount of work in theoretical understanding of their convergence rate under different sets of assumptions. Specifically, recently the convergence properties of EG and OGDA were investigated for SC-SC and C-C settings, where it has been shown that they tend to converge significantly faster than GDA in both deterministic and stochastic settings [39, 12, 40]. Despite these remarkable advances, there is a dearth of theoretical understanding of the convergence of OGDA and EG methods in the nonconvex setting. This naturally motivates us to rigorously examine the convergence of these methods in nonconvex minimax optimization that we aim to investigate. Thus, we emphasize that our focus is on vanilla variants of OGDA/EG, and improved rates in NC-C and NC-SC problems have already been obtained with novel algorithms as mentioned in Section 2.

**Contributions.** We propose a unified framework for analyzing and establishing the convergence of OGDA and EG methods for solving NC-SC and NC-C minimax problems. To the best of our knowledge, our analysis provides the first theoretical guarantees for such problems. Our contribution can be summarized as follows:

- For NC-SC objectives, we demonstrate that OGDA and EG iterates converge to the $\epsilon-$stationary point, with a gradient complexity of $O(\frac{\kappa^2}{\epsilon^2})$ for deterministic case, and $O(\frac{\kappa^3}{\epsilon^4})$ for the stochastic setting, matching the gradient complexity of GDA in [30].

- For NC-C objectives, we establish the gradient complexity of $O(\epsilon^{-6})$ for the deterministic and $O(\epsilon^{-8})$ for stochastic oracles, respectively. Compared to the most analogous work on GDA [30], our rate matches the gradient complexity of GDA our results show that OGDA and EG have the advantage of shaving off a significant term related to primal function gap $(\hat{\Delta}_0 = \Phi(\boldsymbol{x}_0) - \min_{\boldsymbol{x}} \Phi(\boldsymbol{x}))$.

- We establish impossibility results on the achievable rates by providing an $\Omega(\frac{\kappa^2}{\epsilon^2})$, and $\Omega(\epsilon^{-6})$ lower bounds based on the common choice of parameters for both OGDA and EG in deterministic NC-SC and NC-C settings, respectively, thus demonstrating the tightness of our analysis of upper bounds.

- By carefully designing hard instances, we establish a general lower bound of $O(\frac{\kappa}{\epsilon^2})$, independent of the learning rate, for GDA/OGDA/EG methods in deterministic NC-SC setting–demonstrating the optimality of obtained upper bound up to a factor of $\kappa$.

## 2 Related Work

**Extra-gradient (EG), and OGDA methods.** Under smooth SC-SC assumption, deterministic OGDA and EG have been shown to converge to an $O(\epsilon)$ neighborhood of the optimal solution with rate of $O(\kappa \log(\frac{1}{\epsilon}))$ [39, 49]. Fallah et al. [12] improved upon the previous rates by proposing multistage

OGDA, which achieved the best-known rate of $O(\max(\kappa \log(\frac{1}{\epsilon}), \frac{\sigma^2}{\mu^2 \epsilon^2}))$ for the stochastic OGDA in SC-SC setting. Under monotone and gradient Lipschitzness assumption (a slightly weaker notion of smooth convex-concave problems), Cai et al. [3] established the tight last iterate convergence of $O(\frac{1}{\sqrt{T}})$ for OGDA and EG, and similar results for EG has been achieved in [17, 16]. Furthermore, To the best of our knowledge, OGDA and EG methods have not been extensively explored in nonconvex-nonconcave settings except in a few recent works on structured nonconvex-nonconcave problems in which the analysis is done through the lens of a variational inequality. This line of work is discussed in the Nonconvex-nonconcave section. Moreover, recently, Guo et al. [18] established the convergence rate of OGDA in NC-SC, however, they have $\mu$-PL assumption on $\Phi(\boldsymbol{x})$, which is a strong assumption and further allows them to show the convergence rate in terms of the objective gap. However, we did not make such an assumption on the primal function, and hence unlike [18], we measure the convergence by the gradient norm of the primal function.

**Nonconvex-strongly-concave (NC-SC) problems.** In deterministic setting, Lin et al. [30] demonstrated the first non-asymptotic convergence of GDA to $\epsilon$-stationary point of $\Phi(\boldsymbol{x})$, with the gradient complexity of $O(\frac{\kappa^2}{\epsilon^2})$. Lin et al. [31] and Zhang et al. [55] proposed triple loop algorithms achieving gradient complexity of $O(\frac{\sqrt{\kappa}}{\epsilon^2})$ by leveraging ideas from catalyst methods (adding $\alpha\|\boldsymbol{x} - \boldsymbol{x}_0\|^2$ to the objective function), and inexact proximal point methods, which nearly match the existing lower bound [27, 55, 20]. Approximating the inner loop optimization of catalyst idea by one step of GDA, Yang et al [52] developed a single loop algorithm called smoothed AGDA, which provably converges to $\epsilon$-stationary point, with gradient complexity of $O(\frac{\kappa}{\epsilon^2})$. For stochastic setting, Lin et al [30] showed that Stochastic GDA, with choosing dual and primal learning rate ratio of $O(\frac{1}{\kappa^2})$, converges to $\epsilon$-stationary point with gradient complexity of $O(\frac{\kappa^3}{\epsilon^4})$. Chen et al. [4] proposed a double loop algorithm whose outer loop performs one step of gradient descent on the primal variable, and inner loop performs multiple steps of gradient ascent. Using this idea, they achieved gradient complexity of $O(\frac{\kappa^3}{\epsilon^4})$ with fixed batch size. However, their algorithm is double loop, and the iteration complexity of the inner loop is $O(\kappa)$. Yang et al [52] also introduced the stochastic version of smoothed AGDA we mentioned earlier. They showed gradient complexity of $O(\frac{\kappa^2}{\epsilon^4})$, using fixed batch size. They achieved the best-known rate for NC-PL problems, which is an even weaker assumption than NC-SC.

**Nonconvex-concave.** Recently, due to the surge of GANs [14] and adversarially robust neural network training, a line of researches are focusing on nonconvex-concave or even nonconvex-nonconcave minimax optimization problems [36, 29, 41, 44, 48, 13, 32, 33, 24]. For nonconvex-concave setting, to our best knowledge, Rafique et al [44] is the pioneer to propose provable nonconvex-concave minimax algorithm, where they proposed Proximally Guided Stochastic Mirror Descent Method, which achieves $O(\epsilon^{-6})$ gradient complexity to find stationary point. Nouiehed et al [41] presented a double-loop algorithm to solve nonconvex-concave with constraint on both $\boldsymbol{x}$ and $\boldsymbol{y}$, and achieved $O(\epsilon^{-7})$ rate. Lin et al [30] provided the first analysis of the classic algorithm (S)GDA on nonconvex-strongly-concave and nonconvex-concave functions, and in nonconvex-concave setting they achieve $O(\epsilon^{-6})$ for GDA and $O(\epsilon^{-8})$ for SGDA. Zhang et al [53] proposed smoothed-GDA and also achieve $O(\epsilon^{-8})$ rate. Thekumparampil et al. [48] proposed Proximal Dual Implicit Accelerated Gradient method and achieved the best known rate $O(\epsilon^{-3})$ for nonconvex-concave problem. Kong and Monteiro [26] proposed an accelerated inexact proximal point method and also achieve $O(\epsilon^{-3})$ rate. Lin et al [31] designed near-optimal algorithm using an acceleration method with $O(\epsilon^{-3})$ rate. However, their algorithms require double or triple loops and are not as easy to implement as GDA, OGDA, or EG methods.

**Nonconvex-nonconcave.** Minimax optimization problems can be cast as one of the special cases of variational inequality problems (VIPs) [1, 34]. Thus, one way of studying the convergence in Nonconvex-nonconcave problems is to leverage some variants of Variational Inequality properties such as Monotone variational inequality, Minty variational inequality (MVI), weak MVI, and negative comonotone, which are weaker assumptions compared to convex-concave problems. For instance, Loizou et al. [35] showed the linear convergence of SGDA under expected co-coercivity, a condition that potentially holds for the non-monotone problem. Moreover, it has been shown that deterministic EG obtains gradient complexity of $O(\frac{1}{\epsilon^2})$ for the aforementioned settings [7, 10, 47, 42]. Alternatively, another line of works established the convergence under the weaker notions of strong convexity such as the Polyak-Łojasiewicz (PL) condition, or $\rho$-weakly convex. Yang et al [51] established the linear convergence of the AGDA algorithm assuming the two-sided PL condition. Hajizadeh et al [19] achieved the same results for EG under the weakly-convex, weakly-concave assumption.

# 3 Problem setup and preliminaries

We use lower-case boldface letters such as $\boldsymbol{x}$ to denote vectors and let $\|\cdot\|$ denote the $\ell_2$-norm of vectors. In Problem 1, we refer to $\boldsymbol{x}$ as the primal variable and to $\boldsymbol{y}$ as the dual variable. For a function $f: \mathbb{R}^m \times \mathbb{R}^n \to \mathbb{R}$, we use $\nabla_x f(\boldsymbol{x}, \boldsymbol{y})$ to denote the gradient of $f(\boldsymbol{x}, \boldsymbol{y})$ with respect to primal variable $\boldsymbol{x}$, and $\nabla_y f(\boldsymbol{x}, \boldsymbol{y})$ to denote the gradient of $f(\boldsymbol{x}, \boldsymbol{y})$ with respect to dual variable $\boldsymbol{y}$. In stochastic setting, we let $\boldsymbol{g}_{x,t}$ to be the unbiased estimator of $\nabla_x f(\boldsymbol{x}_t, \boldsymbol{y}_t)$, computed by a minibatch of size $M_x$ and $\boldsymbol{g}_{y,t}$ to be the unbiased estimator of $\nabla_y f(\boldsymbol{x}_t, \boldsymbol{y}_t)$, computed by a minibatch of size $M_y$, where $\boldsymbol{x}_t$ and $\boldsymbol{y}_t$ are the $t$th iterates of the algorithms. Particularly, $\boldsymbol{g}_{x,t} = \frac{1}{M_x} \sum_{i=1}^{M_x} \nabla_x f(\boldsymbol{x}_t, \boldsymbol{y}_t, \xi_{t,i}^x)$, and $\boldsymbol{g}_{y,t} = \frac{1}{M_y} \sum_{i=1}^{M_y} \nabla_y f(\boldsymbol{x}_t, \boldsymbol{y}_t, \xi_{t,i}^y)$, where $\{\xi_{t,i}^x\}_{i=1}^{M_x}$, and $\{\xi_{t,i}^y\}_{i=1}^{M_y}$ are i.i.d minibatch samples utilized to compute stochastic gradients at each iteration $t \in \{1, \ldots, T\}$.

**Definition 3.1** (Primal Function). We introduce $\Phi(\boldsymbol{x}) = \max_{\boldsymbol{y}} f(\boldsymbol{x}, \boldsymbol{y})$ as the primal function, and define $\boldsymbol{y}^*(\boldsymbol{x}) = \arg\max_{\boldsymbol{y} \in \mathcal{Y}} f(\boldsymbol{x}, \boldsymbol{y})$ as the optimal dual variable at a point $\boldsymbol{x}$.

**Definition 3.2** (Smoothness). A function $f(\boldsymbol{x}, \boldsymbol{y})$ is $\ell$-smooth in both $\boldsymbol{x}$, and $\boldsymbol{y}$, if it is differentiable, and the following inequalities hold: $\|\nabla f(\boldsymbol{x}_1, \boldsymbol{y}_1) - \nabla f(\boldsymbol{x}_2, \boldsymbol{y}_2)\|^2 \leq \ell^2 \|\boldsymbol{x}_1 - \boldsymbol{x}_2\|^2 + \ell^2 \|\boldsymbol{y}_1 - \boldsymbol{y}_2\|^2$.

**Definition 3.3.** A function $g$ is $\mu$-strongly-convex, if for any $\boldsymbol{x}_1, \boldsymbol{x}_2 \in \mathbb{R}^d$ the following holds: $g(\boldsymbol{x}_2) \geq g(\boldsymbol{x}_1) + \langle \nabla g(\boldsymbol{x}_1), \boldsymbol{x}_2 - \boldsymbol{x}_1 \rangle + \frac{\mu}{2} \|\boldsymbol{x}_1 - \boldsymbol{x}_2\|^2$.

**Definition 3.4.** We say $\boldsymbol{x}$ is is an $\epsilon$-stationary point for a differentiable function $\Phi$ if $\|\nabla \Phi(\boldsymbol{x})\| \leq \epsilon$.

We note that $\epsilon$-stationary point is a common optimality criterion used in the NC-SC setting. As pointed out in [30], considering $\Phi(\boldsymbol{x})$ as convergence measure is natural since in many application scenarios, we mainly care about the value of the objective $f(\boldsymbol{x}, \boldsymbol{y})$ under the maximized $\boldsymbol{y}$, e.g., adversarial training or distributionally robust learning.

When $f(\boldsymbol{x}, \boldsymbol{y})$ is merely concave in $\boldsymbol{y}$, $\Phi(\boldsymbol{x})$ could be non-differentiable. Hence, following the routine of nonsmooth nonconvex minimization [9], we consider the following Moreau envelope function:

**Definition 3.5** (Moreau envelope). A function $\Phi_p(\boldsymbol{x})$ is the $p$-Moreau envelope of a function $\Phi$ if $\Phi_p(\boldsymbol{x}) := \min_{\boldsymbol{x}' \in \mathbb{R}^d} \{\Phi(\boldsymbol{x}') + \frac{1}{2p} \|\boldsymbol{x}' - \boldsymbol{x}\|^2\}$.

We will utilize the following property of the Moreau envelope of a nonsmooth function:

**Lemma 3.6** (Davis and Drusvyatskiy [9]). *Let $\hat{\boldsymbol{x}} = \arg\min_{\boldsymbol{x}' \in \mathbb{R}^d} \Phi(\boldsymbol{x}') + \frac{1}{2p} \|\boldsymbol{x}' - \boldsymbol{x}\|^2$, then the following inequalities hold: $\|\hat{\boldsymbol{x}} - \boldsymbol{x}\| \leq p \|\nabla \Phi_p(\boldsymbol{x})\|$, $\min_{\boldsymbol{v} \in \partial \Phi(\hat{\boldsymbol{x}})} \|\boldsymbol{v}\| \leq \|\nabla \Phi_p(\boldsymbol{x})\|$.*

Lemma 3.6 suggests that, if we can find a $\boldsymbol{x}$ with a small $\|\nabla \Phi_p(\boldsymbol{x})\|$, then $\boldsymbol{x}$ is near some point $\hat{\boldsymbol{x}}$ which is a near-stationary point of $\Phi$. We will use $1/2\ell$-Moreau envelope of $\Phi$, following the setting in [30, 45], and establish the convergence rates in terms of $\|\nabla \Phi_{1/2\ell}(\boldsymbol{x})\|$. We also define two quantities $\hat{\Delta}_\Phi = \Phi_{1/2\ell}(\boldsymbol{x}_0) - \min_{\boldsymbol{x} \in \mathbb{R}^d} \Phi_{1/2\ell}(\boldsymbol{x})$ and $\hat{\Delta}_0 = \Phi(\boldsymbol{x}_0) - \min_{\boldsymbol{x} \in \mathbb{R}^d} \Phi(\boldsymbol{x})$ that appear in our convergence bounds. Before presenting our results on EG and OGDA, we briefly revisit the most related algorithm, Gradient Descent Ascent (GDA).

## 3.1 Gradient Descent Ascent (GDA) algorithm

The GDA method, as detailed in Algorithm 1, performs simultaneous gradient descent and ascent updates on primal and dual variables, respectively. This simple algorithm has been deployed extensively for minimax optimization applications such as Generative Adversarial Networks (GANs). Under Assumptions 4.1, and 4.3, Lin et al. [30] established the convergence of GDA by choosing

---

**Algorithm 1** GDA

**Input:** $(\boldsymbol{x}_0, \boldsymbol{y}_0)$, stepsizes $(\eta_x, \eta_y)$
**for** $t = 1, 2, \ldots, T$ **do**
  $\boldsymbol{x}_t \leftarrow \boldsymbol{x}_{t-1} - \eta_x \nabla_x f(\boldsymbol{x}_{t-1}, \boldsymbol{y}_{t-1})$ ;
  $\boldsymbol{y}_t \leftarrow \mathcal{P}_{\mathcal{Y}}(\boldsymbol{y}_{t-1} + \eta_y \nabla_y f(\boldsymbol{x}_{t-1}, \boldsymbol{y}_{t-1}))$ ;
**end for**
Randomly choose $\bar{\boldsymbol{x}}$ from $\boldsymbol{x}_1, \ldots, \boldsymbol{x}_T$
**Output:** $\bar{x}$

---

$\eta_x = \Theta(\frac{1}{\kappa^2 \ell})$, and $\eta_y = \Theta(\frac{1}{\ell})$. In particular, they showed that deterministic GDA requires $O(\frac{\kappa^2}{\epsilon^2})$ calls to a gradient oracle, and stochastic GDA requires $O(\frac{\kappa^3}{\epsilon^4})$ calls using the minibatch size of $O(\frac{\kappa}{\epsilon^2})$ to find an $\epsilon$-stationary point of the primal function.

## 3.2 Optimistic Gradient Descent Ascent (OGDA) and Extra-gradient (EG) Method

We now turn to reviewing the algorithms we study in this paper: Optimistic GDA (OGDA) and Extra-gradient (EG) methods. To optimize Problem (1), at each iteration $t = 1, 2, \ldots, T$, OGDA performs the following updates on the primal and dual variables:

$$
\begin{aligned}
\boldsymbol{x}_{t+1} &= \boldsymbol{x}_t - \eta_x \nabla_x f(\boldsymbol{x}_t, \boldsymbol{y}_t) - \eta_x (\nabla_x f(\boldsymbol{x}_t, \boldsymbol{y}_t) - \nabla_x f(\boldsymbol{x}_{t-1}, \boldsymbol{y}_{t-1})) \\
\boldsymbol{y}_{t+1} &= \mathcal{P}_{\mathcal{Y}} \left( \boldsymbol{y}_t + \eta_y \nabla_y f(\boldsymbol{x}_t, \boldsymbol{y}_t) + \eta_y (\nabla_y f(\boldsymbol{x}_t, \boldsymbol{y}_t) - \nabla_y f(\boldsymbol{x}_{t-1}, \boldsymbol{y}_{t-1})) \right)
\end{aligned}
\tag{OGDA}
$$

where correction terms (e.g. $\nabla_x f(\boldsymbol{x}_t, \boldsymbol{y}_t) - \nabla_x f(\boldsymbol{x}_{t-1}, \boldsymbol{y}_{t-1})$) are added to the updates of the GDA. EG method performs the following updates:

$$
\begin{aligned}
\boldsymbol{x}_{t+1/2} &= \boldsymbol{x}_t - \eta_x \nabla_x f(\boldsymbol{x}_t, \boldsymbol{y}_t) \\
\boldsymbol{x}_{t+1} &= \boldsymbol{x}_t - \eta_x \nabla_x f(\boldsymbol{x}_{t+1/2}, \boldsymbol{y}_{t+1/2})
\end{aligned}
;
\quad
\begin{aligned}
\boldsymbol{y}_{t+1/2} &= \mathcal{P}_{\mathcal{Y}} \left( \boldsymbol{y}_t + \eta_y \nabla_y f(\boldsymbol{x}_t, \boldsymbol{y}_t) \right) \\
\boldsymbol{y}_{t+1} &= \mathcal{P}_{\mathcal{Y}} \left( \boldsymbol{y}_t + \eta_y \nabla_y f(\boldsymbol{x}_{t+1/2}, \boldsymbol{y}_{t+1/2}) \right)
\end{aligned}
\tag{EG}
$$

where the gradient at the current point is used to find a mid-point, and then the gradient at the mid-point is used to find the next iterate. We also consider *stochastic* variants of the two algorithms where we replace full gradients with unbiased stochastic estimations. The detailed versions of these algorithms are provided in Algorithm 2 , and Algorithm 3 in Appendix A.

# 4 Main Results

We provide upper bounds on the gradient complexity and iteration complexity of OGDA and EG methods for NC-C and NC-SC objectives in both deterministic and stochastic settings. We also show the tightness of obtained bounds for the choice of learning rates made. We will derive general stepsize-independent lower bounds in Section 5.

## 4.1 Nonconvex-strongly-concave minimax problems

We start by establishing the convergence of deterministic OGDA/EG in the NC-SC setting by making the following standard assumption on the loss function.

**Assumption 4.1.** We assume $f : \mathbb{R}^m \times \mathbb{R}^n \to \mathbb{R}$ is $\ell$-smooth, and $f(\boldsymbol{x}, .)$ is $\mu$-strongly-concave.

Moreover, we assume the initial primal optimality gap is bounded. i.e., $\Delta_\Phi = \max(\Phi(\boldsymbol{x}_1), \Phi(\boldsymbol{x}_0)) - \min_x \Phi(\boldsymbol{x})$.

**Theorem 4.2.** *Let $\bar{\boldsymbol{x}}$ be output of OGDA/EG algorithms and choose $\eta_x \leq \frac{c_1}{\kappa^2 \ell}$, $\eta_y = \frac{c_2}{\ell}$. For OGDA, let $c_1 = \frac{1}{50}, c_2 = \frac{1}{6}$, and for EG, let $c_1 = \frac{1}{75}, c_2 = \frac{1}{4}$. Then under Assumption 4.1, OGDA/EG converges to an $\epsilon$-stationary point, i.e., $\|\nabla \Phi(\bar{\boldsymbol{x}})\|^2 \leq \epsilon^2$, with iteration number $T$ bounded by:*

$$
O \left( \frac{\kappa^2 \ell \Delta_\Phi + \kappa \ell^2 D_0}{\epsilon^2} \right),
$$

*where $D_0 = \max \left( \|\boldsymbol{x}_1 - \boldsymbol{x}_0\|^2, \|\boldsymbol{y}_1 - \boldsymbol{y}_0\|^2, \|\boldsymbol{y}_1 - \boldsymbol{y}_1^*\|^2, \|\boldsymbol{y}_0 - \boldsymbol{y}_0^*\|^2 \right)$.*

To establish the convergence rate in stochastic setting, we will make the following assumption on the stochastic gradient oracle.

**Assumption 4.3.** Let $\nabla_x f(\boldsymbol{x}, \boldsymbol{y}, \xi^x)$ and $\nabla_y f(\boldsymbol{x}, \boldsymbol{y}, \xi^y)$ to be the unbiased estimator of the $\nabla_x f(\boldsymbol{x}, \boldsymbol{y})$ and $\nabla_y f(\boldsymbol{x}, \boldsymbol{y})$, respectively. Then, the stochastic gradient oracle satisfies the following:

- Unbiasedness: $\mathbb{E}_{\xi^x} [\nabla_x f(\boldsymbol{x}, \boldsymbol{y}, \xi^x)] = \nabla_x f(\boldsymbol{x}, \boldsymbol{y})$ and $\mathbb{E}_{\xi^y} [\nabla_y f(\boldsymbol{x}, \boldsymbol{y}, \xi^y)] = \nabla_y f(\boldsymbol{x}, \boldsymbol{y})$.
- Bounded variance: We assume the variance of stochastic gradients are bounded, i.e., $\mathbb{E}_{\xi^x} [\|\nabla_x f(\boldsymbol{x}, \boldsymbol{y}, \xi^x) - \nabla_x f(\boldsymbol{x}, \boldsymbol{y})\|^2] \leq \sigma^2$ and $\mathbb{E}_{\xi^y} [\|\nabla_y f(\boldsymbol{x}, \boldsymbol{y}, \xi^y) - \nabla_y f(\boldsymbol{x}, \boldsymbol{y})\|^2] \leq \sigma^2$.

We now turn to establishing the convergence rate in stochastic setting.

**Theorem 4.4.** *Let $\bar{\boldsymbol{x}}$ be output of stochastic OGDA/EG algorithms and let $\eta_x$ and $\eta_y$ to be chosen as in Theorem 4.2. For EG, choose minibatch size $M = \max \left\{ 1, \frac{\kappa \sigma^2}{\epsilon^2} \right\}$, and for OGDA choose*

*primal minibatch size $M_x = \max\{1, \frac{\sigma^2}{\epsilon^2}\}$, and dual minibatch size $M_y = \max\{1, \frac{\kappa\sigma^2}{\epsilon^2}\}$. Then under Assumptions 4.1, and 4.3, OGDA/EG converges to an $\epsilon$-stationary point, i.e., $\mathbb{E}\|\nabla\Phi(\bar{\boldsymbol{x}})\|^2 \leq \epsilon^2$, with the iteration number $T$ bounded by:*

$$O\left(\frac{\kappa^2\ell\Delta_\Phi + \kappa\ell^2 D_0}{\epsilon^2}\right),$$

*where $D_0 = \max\left(\|\boldsymbol{x}_1 - \boldsymbol{x}_0\|^2, \|\boldsymbol{y}_1 - \boldsymbol{y}_0\|^2, \|\boldsymbol{y}_1 - \boldsymbol{y}_1^*\|^2, \|\boldsymbol{y}_0 - \boldsymbol{y}_0^*\|^2\right).$*

The proofs of Theorems 4.2 and 4.4 are deferred to Appendix A. Our iteration complexity matches with the complexity of two-scale GDA obtained in [30]. However, we improve primal gradient oracle complexity for OGDA by a factor of $\kappa$ as our analysis works for smaller primal batch size $M_x$ compared to GDA [30]. This paper establishes primal gradient oracle complexity of $O(\frac{\kappa^2}{\epsilon^4})$, while the analysis for GDA in [30], requires gradient oracle complexity of $O(\frac{\kappa^3}{\epsilon^4})$ for primal variable.

In previous theorems, we established upper bounds on the convergence of OGDA and EG algorithms. In the following results, we turn to examining the tightness of obtained rates. To this end, we first consider a simple GDA algorithm and will extend the analysis to OGDA/EG. Note that in this section, we only consider the stepsize choice in our upper bound results.

**Theorem 4.5** (Tightness of GDA). *Consider GDA method (Algorithm 1) with step sizes chosen as in Theorem 4.4 in [30], and let $\bar{\boldsymbol{x}}$ be the returned solution after $T$ iterations. Then, there exists a function $f(\cdot, \cdot)$ that is $\ell$-gradient Lipschitz and $\mu$-strongly concave in $\boldsymbol{y}$, and an initialization $(\boldsymbol{x}_0, \boldsymbol{y}_0)$, such that Algorithm 1 requires at least $T = \Omega\left(\frac{\kappa^2\Delta_\Phi}{\epsilon^2}\right)$ iterations to guarantee $\|\nabla\Phi(\bar{\boldsymbol{x}})\| \leq \epsilon$.*

**Theorem 4.6** (Tightness of EG/OGDA). *Consider deterministic EG and OGDA methods with step sizes chosen as in Theorem 4.2 and let $\bar{\boldsymbol{x}}$ be the returned solution after $T$ iterations. Then, there exists a function $f(\cdot, \cdot)$ that is $\ell$-gradient Lipschitz and $\mu$-strongly concave in $\boldsymbol{y}$, and an initialization $(\boldsymbol{x}_0, \boldsymbol{y}_0)$, such that both methods require at least $T = \Omega\left(\frac{\kappa^2\Delta_\Phi}{\epsilon^2}\right)$ iterations to guarantee $\|\nabla\Phi(\bar{\boldsymbol{x}})\| \leq \epsilon$.*

The proofs of Theorems 4.5 and 4.6 are deferred to Appendix A.3.1 and A.3.2, respectively. Theorems 4.6 show that to achieve $\epsilon$ stationary point of $\Phi$, EG and OGDA need at least $O(\frac{\kappa^2}{\epsilon^2})$ gradient evaluations, which match with our upper bound results (Theorems 4.2). These impossibility results demonstrate the tightness of our analysis. It would also be interesting to see such analysis for stochastic setting, which we leave as a valuable future work.

### 4.2 Nonconvex-concave minimax problems

We now turn to establishing the convergence rate of (stochastic) OGDA/EG in the NC-C setting. We make the following assumption throughout this subsection:

**Assumption 4.7.** We assume $f : \mathbb{R}^m \times \mathcal{Y} \to \mathbb{R}$ is $\ell$-smooth in $\boldsymbol{x}, \boldsymbol{y}$, $G$-Lipschitz in $\boldsymbol{x}$ and $\mathcal{Y}$ is bounded convex set with diameter $D$, and also $f(\boldsymbol{x}, .)$ is concave.

From the above assumption, we note when $f$ is merely concave in $\boldsymbol{y}$, we have to assume the dual variable domain is bounded since otherwise, the Moreau envelope function will not be well-defined (This is shown in Lemma 3.6 in [30]). Therefore, the update rule for $\boldsymbol{y}$ requires projection as follows:

$$\boldsymbol{y}_t = \mathcal{P}_{\mathcal{Y}}\left(\boldsymbol{y}_{t-1} + \eta_y\nabla_y f(\boldsymbol{x}_{t-1}, \boldsymbol{y}_{t-1}) + \eta_y(\nabla_y f(\boldsymbol{x}_{t-1}, \boldsymbol{y}_{t-1}) - \nabla_y f(\boldsymbol{x}_{t-2}, \boldsymbol{y}_{t-2})))\right) \quad \text{(OGDA)}$$

$$\boldsymbol{y}_{t+1/2} = \mathcal{P}_{\mathcal{Y}}\left(\boldsymbol{y}_t + \eta_y\nabla_y f(\boldsymbol{x}_t, \boldsymbol{y}_t)\right), \quad \boldsymbol{y}_{t+1} = \mathcal{P}_{\mathcal{Y}}\left(\boldsymbol{y}_t + \eta_y\nabla_y f(\boldsymbol{x}_{t+1/2}, \boldsymbol{y}_{t+1/2})\right) \quad \text{(EG)}$$

The following theorem establishes the convergence of OGDA/EG for NC-C objectives.

**Theorem 4.8.** *Let $\eta_x = O\left(\min\left\{\frac{\epsilon}{\ell G}, \frac{\epsilon^2}{\ell G^2}, \frac{\epsilon^4}{D^2 G^2\ell^3}\right\}\right)$, and $\eta_y = \frac{1}{2\ell}$. By convention, we set $\boldsymbol{x}_{-1/2} = \boldsymbol{x}_0$, $\boldsymbol{y}_{-1/2} = \boldsymbol{y}_0$. Under Assumption 4.7, OGDA/EG converges to an $\epsilon$-stationary point, i.e., $\frac{1}{T+1}\sum_{t=0}^{T}\|\nabla\Phi_{1/2\ell}(\boldsymbol{x}_t)\|^2 \leq \epsilon^2$ for OGDA and $\frac{1}{T+1}\sum_{t=0}^{T}\|\nabla\Phi_{1/2\ell}(\boldsymbol{x}_{t-1/2})\|^2 \leq \epsilon^2$ for EG, with the gradient complexity bounded by:*

$$O\left(\frac{\ell G^2\hat{\Delta}_\Phi}{\epsilon^4}\max\left\{1, \frac{D^2\ell^2}{\epsilon^2}\right\}\right).$$

**Theorem 4.9.** *Let* $\eta_x = O(\min\{\frac{\epsilon^2}{\ell(G^2+\sigma^2)}, \frac{\epsilon^4}{D^2\ell^3 G\sqrt{G^2+\sigma^2}}, \frac{\epsilon^6}{D^2\ell^3\sigma^2 G\sqrt{G^2+\sigma^2}}\})$, *and* $\eta_y = O(\min\{\frac{1}{4\ell}, \frac{\epsilon^2}{\ell\sigma^2}\})$. *By convention, we set* $\boldsymbol{x}_{-1/2} = \boldsymbol{x}_0$, $\boldsymbol{y}_{-1/2} = \boldsymbol{y}_0$. *Under Assumptions 4.3 and 4.7, stochastic OGDA/EG algorithms converge to an $\epsilon$-stationary point, i.e.,* $\frac{1}{T+1}\sum_{t=0}^T \mathbb{E}\|\nabla\Phi_{1/2\ell}(\boldsymbol{x}_t)\|^2 \le \epsilon^2$ *for OGDA and* $\frac{1}{T+1}\sum_{t=0}^T \mathbb{E}\|\nabla\Phi_{1/2\ell}(\boldsymbol{x}_{t-1/2})\|^2 \le \epsilon^2$ *for EG, with the gradient complexity bounded by:*

$$O\left(\frac{D^2\ell^3 G\sqrt{G^2+\sigma^2}\hat{\Delta}_\Phi}{\epsilon^6}\max\left\{1, \frac{\sigma^2}{\epsilon^2}\right\}\right).$$

The proofs of Theorems 4.8 and 4.9 are deferred to Appendix B. Here we show that OGDA/EG need at most $O\left(\frac{D^2\ell^3 G^2\hat{\Delta}_\Phi}{\epsilon^6}\right)$ gradient evaluations in deterministic setting and $O\left(\frac{D^2\ell^3\sigma^2 G\sqrt{G^2+\sigma^2}\hat{\Delta}_\Phi}{\epsilon^8}\right)$ gradient evaluations in stochastic setting to visit an $\epsilon$-stationary point.

Our stepsize choices for dual variable match the optimal analysis in convex-concave setting, $\Theta(\frac{1}{\ell})$ in deterministic setting [40] and $\Theta(\frac{1}{\epsilon^2})$ in stochastic setting [23], so we suppose our dual stepsize choice is optimal. The stepsize ratio is $\frac{\eta_x}{\eta_y} = O(\epsilon^4)$ in both settings, same as Lin et al. [30]'s results on applying GDA to a nonconvex-concave objective, which reveals some connection and similarity between OGDA and GDA. However, compared to GDA [30], where they get an $O\left(\frac{D^2\ell^3 G^2\hat{\Delta}_\Phi}{\epsilon^6} + \frac{\ell^3 D^2\hat{\Delta}_0}{\epsilon^4}\right)$ rate in deterministic setting, and $O\left(\frac{D^2\ell^3\sigma^2 G\sqrt{G^2+\sigma^2}\hat{\Delta}_\Phi}{\epsilon^8} + \frac{\ell^3 D^2\hat{\Delta}_0}{\epsilon^6}\right)$ in stochastic setting, we shave off the significant terms with dependency on $\hat{\Delta}_0$. As we will show in the proof, this acceleration is mainly due to the fact that OGDA/EG enjoys an inherent nice descent property on concave function, which is more elaborated in Section 4.3. In the stochastic setting, we observe similar superiority.

Now, we switch to examining the tightness of obtained rates. Similar to the NC-SC setting, we first consider a simple GDA algorithm and will extend the analysis to OGDA/EG.

**Theorem 4.10** (Tightness of GDA ). *Consider GDA that runs $T$ iterations on solving (1), and let $\boldsymbol{x}_T$ be the returned solution. Then, there exists a function $f$ that is $G$-Lipschitz in $\boldsymbol{x}$, $\ell$-gradient Lipschitz and concave in $\boldsymbol{y}$, and an initialization point $(\boldsymbol{x}_0, \boldsymbol{y}_0)$ such that GDA requires at least $T = \Omega\left(\frac{\ell^3 G^2 D^2\hat{\Delta}_\Phi}{\epsilon^6}\right)$ iterations to guarantee $\|\Phi_{1/2\ell}(\boldsymbol{x}_T)\| \le \epsilon$.*

**Theorem 4.11** (Tightness of OGDA/EG). *Consider OGDA/EG that runs $T$ iterations on solving (1), and let $\boldsymbol{x}_T$ be the returned solution. Then, there exists a function $f$ that is $G$-Lipschitz in $\boldsymbol{x}$, $\ell$-gradient Lipschitz and concave in $\boldsymbol{y}$, and an initialization point $(\boldsymbol{x}_0, \boldsymbol{y}_0)$ such that to achieve $\|\Phi_{1/2\ell}(\boldsymbol{x}_T)\| \le \epsilon$, OGDA/EG requires at least $T = \Omega\left(\frac{\ell^3 G^2 D^2\hat{\Delta}_\Phi}{\epsilon^6}\right)$.*

The proof of Theorems 4.10 and 4.11 are deferred to Appendix B.3.1 and B.3.2, respectively. Theorems 4.11 demonstrates that to find an $\epsilon$ stationary point of $\Phi_{1/2\ell}$, OGDA and EG with our stepsize choices need at least $O(\frac{1}{\epsilon^6})$ gradient evaluations, which verifies the tightness of upper bound.

## 4.3 Discussion

**Key technical challenges.** Here, we present the key technical challenges that arise in the nonconvex setting, which makes the analysis much more involved compared to the previous analysis of these algorithms in convex settings. Our proofs are mainly based on NC-C and NC-SC GDA analysis in [30], and SC-SC OGDA/EG analysis in [39]. In the nonconvex-strongly-concave setting, finding an upper bound for $\sum_{i=1}^T \|\boldsymbol{y}_i - \boldsymbol{y}^*(\boldsymbol{x}_i)\|^2$ is one of the key steps to establish the convergence rate, however bounding this term is much more complicated for OGDA and EG than GDA due to difference in updating rules. Note that in GDA analysis [30], $\sum_{i=1}^T \|\boldsymbol{y}_i - \boldsymbol{y}^*(\boldsymbol{x}_i)\|^2$ can be bounded by deriving simple recursive equation for $\|\boldsymbol{y}_t - \boldsymbol{y}^*(\boldsymbol{x}_t)\|^2$, while extending it to OGDA is quite complicated. Hence, we propose to bound $r_t = \|\boldsymbol{z}_{t+1} - \boldsymbol{y}^*(\boldsymbol{x}_t)\|^2 + \frac{1}{4}\|\boldsymbol{y}_t - \boldsymbol{y}_{t-1}\|^2$, and establish the upper bound on $\sum_{i=1}^t \|\boldsymbol{y}_i - \boldsymbol{y}^*(\boldsymbol{x}_i)\|^2$ in terms of $\sum_{i=1}^t r_i$. In nonconvex-concave setting, we have to bound $\|\boldsymbol{y}_t - \boldsymbol{y}_{t-1}\|^2$, so we reduce it to the primal function gap: $\Phi(\boldsymbol{x}_t) - f(\boldsymbol{x}_t, \boldsymbol{y}_t)$. To bound this gap, we utilize the benign descent property of OGDA and EG on concave function and shave off a significant term $\hat{\Delta}_0$, which yields a better upper complexity bound than GDA.

**On descent property of concave function for OGDA/EG** Take OGDA, for example. The key step in NC-C analysis is to bound $\Phi(\boldsymbol{x}_t) - f(\boldsymbol{x}_t, \boldsymbol{y}_t)$. In OGDA proof, we split this into bounding the following:

$$
\begin{aligned}
\Phi(\boldsymbol{x}_t) - f(\boldsymbol{x}_t, \boldsymbol{y}_t) \leq \; & f(\boldsymbol{x}_t, \boldsymbol{y}^*(\boldsymbol{x}_t)) - f(\boldsymbol{x}_s, \boldsymbol{y}^*(\boldsymbol{x}_t)) + f(\boldsymbol{x}_s, \boldsymbol{y}^*(\boldsymbol{x}_s)) \\
& - f(\boldsymbol{x}_t, \boldsymbol{y}^*(\boldsymbol{x}_s)) + f(\boldsymbol{x}_t, \boldsymbol{y}^*(\boldsymbol{x}_s)) - f(\boldsymbol{x}_t, \boldsymbol{y}_t).
\end{aligned}
\tag{2}
$$

For the last term $f(\boldsymbol{x}_t, \boldsymbol{y}^*(\boldsymbol{x}_s)) - f(\boldsymbol{x}_t, \boldsymbol{y}_t)$, OGDA can guarantee its convergence without bounded gradient assumption on $\boldsymbol{y}$. However, for GDA, it requires bounded gradient assumption on $\boldsymbol{y}$ to show the convergence of this term, and without such assumption, we can only show the convergence of $f(\boldsymbol{x}_t, \boldsymbol{y}^*(\boldsymbol{x}_s)) - f(\boldsymbol{x}_t, \boldsymbol{y}_{t+1})$, so Lin et al. [30] split the $\Phi(\boldsymbol{x}_t) - f(\boldsymbol{x}_t, \boldsymbol{y}_t)$ as follow:

$$
\begin{aligned}
\Phi(\boldsymbol{x}_t) - f(\boldsymbol{x}_t, \boldsymbol{y}_t) \leq \; & f(\boldsymbol{x}_t, \boldsymbol{y}^*(\boldsymbol{x}_t)) - f(\boldsymbol{x}_t, \boldsymbol{y}^*(\boldsymbol{x}_s)) + f(\boldsymbol{x}_{t+1}, \boldsymbol{y}_{t+1}) - f(\boldsymbol{x}_t, \boldsymbol{y}_t) + f(\boldsymbol{x}_t, \boldsymbol{y}_{t+1}) \\
& - f(\boldsymbol{x}_{t+1}, \boldsymbol{y}_{t+1}) + f(\boldsymbol{x}_t, \boldsymbol{y}^*(\boldsymbol{x}_s)) - f(\boldsymbol{x}_t, \boldsymbol{y}_{t+1})
\end{aligned}
\tag{3}
$$

Hence they reduce the problem to bounding $f(\boldsymbol{x}_t, \boldsymbol{y}^*(\boldsymbol{x}_s)) - f(\boldsymbol{x}_t, \boldsymbol{y}_{t+1})$. Therefore, they have to pay the price for the extra term $f(\boldsymbol{x}_{t+1}, \boldsymbol{y}_{t+1}) - f(\boldsymbol{x}_t, \boldsymbol{y}_t)$.

**Generalized OGDA.** Generalized OGDA algorithm is a variant of OGDA in which different learning rates are used for current gradient $\nabla f(\boldsymbol{x}_t, \boldsymbol{y}_t)$, and the correction term $\nabla f(\boldsymbol{x}_t, \boldsymbol{y}_t) - \nabla f(\boldsymbol{x}_{t-1}, \boldsymbol{y}_{t-1})$. The update rule for this algorithm is as follows:

$$
\begin{aligned}
\boldsymbol{x}_{t+1} &= \boldsymbol{x}_t - \eta_{x,1} \nabla_x f(\boldsymbol{x}_t, \boldsymbol{y}_t) - \eta_{x,2}(\nabla_x f(\boldsymbol{x}_t, \boldsymbol{y}_t) - \nabla_x f(\boldsymbol{x}_{t-1}, \boldsymbol{y}_{t-1})) \\
\boldsymbol{y}_{t+1} &= \mathcal{P}_{\mathcal{Y}} \left( \boldsymbol{y}_t + \eta_{y,1} \nabla_y f(\boldsymbol{x}_t, \boldsymbol{y}_t) + \eta_{y,2}(\nabla_y f(\boldsymbol{x}_t, \boldsymbol{y}_t) - \nabla_y f(\boldsymbol{x}_{t-1}, \boldsymbol{y}_{t-1})) \right)
\end{aligned}
\tag{OGDA+}
$$

Mokhtari et al. [39] introduced this algorithm and established the convergence bound for the bilinear setting while analysis beyond this setting remained as an open problem. In Appendix D, we show that our analysis can be adapted to establish the convergence of the generalized OGDA algorithm. In Section 6, the empirical advantage of generalized OGDA over the state of art optimization algorithms is shown, and it seems this algorithm is a better alternative to OGDA in practice. We also define the correction term ratios $\beta_1 = \frac{\eta_{x,2}}{\eta_{x,1}}$, $\beta_2 = \frac{\eta_{y,2}}{\eta_{y,1}}$, and empirically study the effect of these parameters on convergence. Note that if $\beta_1 = \beta_2 = 1$, generalized OGDA would be same as OGDA. It would also be an interesting future direction to analyze this algorithm for C-C and SC-SC problems to understand its superior performance better.

**Projected OGDA/EG for NC-SC.** Here, we highlight that while our analysis for NC-SC assumes that $\mathcal{Y} = \mathbb{R}^n$, it can be easily extended to a constrained setting, where the dual update is performed under projection onto a convex bounded set $\mathcal{Y}$. In the following, we provide a proof sketch for extending our analysis of OGDA to its projected variant, in which we do the same primal update as unconstrained OGDA and a projected (Optimistic gradient) OG update, as defined in [23], on the dual variable. The main idea behind our dual descent lemma, Lemma A.6, is interpreting OGDA as an extension of the PEG/OG method and then using Theorem 5 of [23] for PEG/OG analysis, which already considers the projected gradient updates. Thus, our Lemma A.6 could be immediately adapted to the projected update. Lemma A.5 can also be extended to projected setting by leveraging Lemma A.1 in [23]. Combining the projected variant of the mentioned lemmas, the convergence could be easily established for projected OGDA/EG.

## 5   Stepsize-Independent Lower Bounds

So far, we have established upper bounds and tightness results given specific stepsize choices. In this section, we turn to establishing general stepsize-independent lower bound results in the NC-SC setting.

**Theorem 5.1** (Lower complexity bound for GDA). *Consider deterministic GDA method (Algorithm 1) with any arbitrary choice of learning rates, and let $\bar{\boldsymbol{x}}$ be the returned solution. Then, there exists a function $f$ satisfying Assumption 4.1, and an initialization $(\boldsymbol{x}_0, \boldsymbol{y}_0)$, such that Algorithm 1 requires at least $T = \Omega\left(\frac{\kappa}{\epsilon^2}\right)$ iterations to guarantee $\|\nabla \Phi(\bar{\boldsymbol{x}})\| \leq \epsilon$.*

Theorem 5.1 implies that GDA algorithm can not find $\epsilon$ stationary point of NC-SC problem with less than with $\Omega(\frac{\kappa}{\epsilon^2})$ many gradient evaluations. This result provides the first known lower bound for the

GDA algorithm in NC-SC, showing that the rate obtained in [30] for the convergence of GDA is tight up to a factor of $\kappa$. The general proof idea is to consider the following quadratic NC-SC function $f : \mathbb{R} \times \mathbb{R} \mapsto \mathbb{R}$, which is strongly-concave in both $x$ and $y$:

$$f(x, y) := -\tfrac{1}{2}\ell x^2 + bxy - \tfrac{1}{2}\mu y^2.$$

By construction, $f$ is nonconvex in $x$ (it is actually concave in $x$) and $\mu$-strongly-concave in $y$. Assume $\kappa := \ell/\mu \geq 4$ and choose $b = \sqrt{\mu(\ell + \mu_x)}$ for some $0 < \mu_x \leq \ell/2$ to be chosen later. Then we know $b \leq \ell/2$, and it is easy to verify that $f$ is $\ell$ smooth. Note that the primal function

$$\Phi(x) = \max_y f(x, y) = \tfrac{1}{2}\mu_x x^2$$

is actually strongly convex. This also justifies the symbol for $\mu_x$. We use GDA to find the solution for $\min_x \max_y f(x, y)$. Indeed for this problem, the optimal solution is achieved at the origin. The stepsizes ratio is chosen as $r = \frac{\eta_y}{\eta_x}$ and $\eta_y = \frac{1}{\ell}$ for some numerical constants $c$. Then the GDA update rule can be written as

$$\begin{pmatrix} x_{k+1} \\ y_{k+1} \end{pmatrix} = (\mathbf{I} + \eta_x \mathbf{M}) \cdot \begin{pmatrix} x_k \\ y_k \end{pmatrix}, \quad \mathbf{M} := \begin{pmatrix} \ell & -b \\ rb & -\mu r \end{pmatrix}. \tag{4}$$

Note that (4) is a linear time-invariant system, and due to the simplicity of quadratic form, we are able to track the dynamic of primal and dual variables. By iterating this linear system and analyzing the eigenvalues of the transition matrix, we are able to lower bound the gradient at final iterations.

Now we turn to the extension of the lower bound analysis of GDA to OGDA/EG as stated below.

**Theorem 5.2** (Lower complexity bound for OGDA/EG). *Consider the deterministic OGDA/EG method with any arbitrary choice of learning rates and let $\bar{x}$ be the returned solution. Then, there exists a function $f$ satisfying Assumption 4.1, and an initialization $(\boldsymbol{x}_0, \boldsymbol{y}_0)$, such that OGDA/EG method requires at least $T = \Omega\left(\frac{\kappa \Delta_\Phi}{\epsilon^2}\right)$ iterations to guarantee $\|\nabla\Phi(\bar{x})\| \leq \epsilon$.*

Theorem 5.2 shows that OGDA/EG methods can not find $\epsilon$-stationary point for any choice of learning rates with less than $\Omega(\frac{\kappa}{\epsilon^2})$ gradient evaluations. Given the upper bounds we derived for deterministic OGDA/EG in section 4.1, our result indicates that our upper bounds is tight up to a factor of $\kappa$, however, we highlight that according to Theorem 4.6, given our choice of the learning rate, our upper bound is exactly tight. The complete proof of Theorems 5.1 and 5.2 are deferred to Appendix C.

## 6 Experiments

In this section, we empirically evaluate the performance of the OGDA algorithm. In particular, we follow [52] and optimize Wasserstein GAN (WGAN) on a synthetic dataset generated from a Gaussian distribution. We mainly follow the setting of [52, 34] to conduct our experiment. We consider optimizing the following WGAN loss, where the generator approximates a one-dimensional Gaussian distribution:

$$\min_{w_G} \max_{w_D} \quad \mathbb{E}_{x \sim \mathcal{N}(\mu, \sigma^2)}[D_{w_D}(x)] - \mathbb{E}_{z \sim \mathcal{N}(0,1)}[D_{w_D}(G_{w_G}(z))] - \lambda \|w_D\|^2 \tag{5}$$

Where $w_G$ and $w_D$ correspond to generator and discriminator parameters, respectively. We define discriminator to be $D(x) = \phi_1 x + \phi_2 x^2$, and generator to be a neural network with one hidden layer with 5 neurons with ReLU activation function, same as the setup considered in [52]. We assume that real data comes from a Gaussian $\mathcal{N}(\mu, \sigma^2)$ distribution, and the generator tries to approximate $\mu$ and $\sigma^2$ using a neural network. We set $\mu = 0$, and $\sigma = 0.1$. $\lambda$ is the regularization parameter which we set to $0.001$. Note that $\lambda$ makes the function strongly-concave/concave in terms of discriminator parameters, so the problem becomes NC-SC/NC-C.

Performance of fine-tuned stochastic OGDA is depicted in Figure 1a, in comparison to ADAM [25], RMSprop, SGDA [30], SAGDA [52], and Smooth-SAGDA [52], which are well-known minimax optimization methods. Our evaluation shows that OGDA outperforms all of these methods, supporting the empirical advantage of OGDA as seen in relevant studies [28, 8]. While our theoretical results show that OGDA/EG might not outperform GDA in terms of convergence rate, comparing the empirical result suggests that OGDA might converge faster. In Figure 1c, the evolution of the Wasserstein distance metric during the training has been shown. While GDA and OGDA are

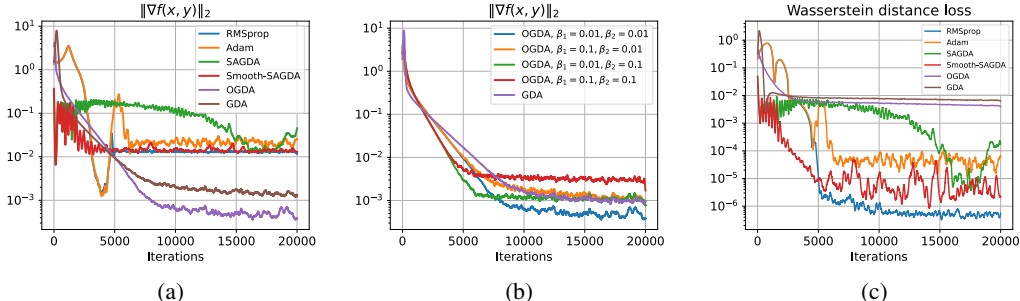

(a)                   (b)                  (c)

Figure 1: Figure 1a demonstrates the best performance of different algorithms on optimizing NC-SC objective in WGAN, where $\|\nabla f(\boldsymbol{x}, \boldsymbol{y})\|^2 = \|\nabla_x f(\boldsymbol{x}, \boldsymbol{y})\|^2 + \|\nabla_y f(\boldsymbol{x}, \boldsymbol{y})\|^2$. For GDA, and OGDA, $\eta_x$, and $\eta_y$ chosen from the set $\{5e-5, 1e-4, 5e-4, 1e-3, 5e-3, 1e-2, 5e-2\}$ using grid search. For OGDA, we choose correction term ratios from the set $\{0, 0.01, 0.1, 0.5, 1\}$. The optimal learning rates are as follows. For both OGDA, and GDA, we set $\eta_x = \eta_y = 0.05$, and for OGDA $\beta_1 = \beta_2 = 0.01$. For other algorithms, we used the same hyperparameters as reported in [52], using the same random seed. Figure 1b indicates effect of tuning correction term ratio $\beta$ on the performance of generalized OGDA algorithm. Figure 1c indicates the evaluation of the Wasserstein distance metric during the training for the best hyperparameter configuration.

stabilized faster than other algorithms, it seems that they converge to a suboptimal solution, which incurs a higher Wasserstein distance. Thus, our study suggests that comparing different minimax algorithms only based on the convergence of gradient norm may not be that insightful in practice, as they might converge to a suboptimal equilibrium. This observation naturally leads to an interesting future direction to theoretically understand how different notions of equilibrium in first-order minimax optimization algorithms are related to the realistic performance of practical methods such as GANs or WGANs.

The common version of OGDA, as depicted in Algorithm 2 in Appendix A, uses the same learning rate for the current gradient and correction term (difference between gradient). Empirically, we observed that using different learning rates for those terms (which we call generalized OGDA) makes the convergence faster and more stable. Hence in the following, we investigate the effect of using different correction term ratios in OGDA, which we refer them as $\beta_1$ and $\beta_2$ as defined in Subsection 4.3. The results in Figure 1b demonstrate that small values of these parameters benefit the convergence rate, and larger values degrade the performance. We further observe that using correction term ratios larger than $0.5$ makes the algorithm diverge and become unstable. Hence, this corroborates the practical importance of the generalized OGDA algorithm compared to OGDA, as we are restricted to choosing the same learning rate in OGDA (i.e., $\beta_1 = \beta_2 = 1$).

## 7 Conclusion

In this paper, we established the convergence of Optimistic Gradient Descent Ascent (OGDA) and Extra-gradient (EG) methods in solving nonconvex minimax optimization problems. We demonstrated that both methods exhibit the same convergence rate that is achievable by GDA in both stochastic and deterministic settings. We also derived matching lower bounds for the choice of parameters that indicate the tightness of obtained rates. Further, we established general lower bounds (i.e, learning rate-independent) for GDA/EG/OGDA in the NC-SC setting, indicating the optimality of obtained upper bounds up to the factor of $\kappa$. It remains an interesting future work to extend the lower bound results to the stochastic setting and also derive the general lower bound for GDA/EG/OGDA in the NC-C setting. Moreover, there is a gap by a factor of $\kappa$ between our lower and upper bounds for NC-SC problems, which would also be an interesting future work to close this gap.

## Acknowledgements

This work was supported in part by NSF grant CNS 1956276. We also would like to thank Mohammad Mahdi Kamani for his help on conducting the experiments.

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
