# OpenReview forum: "Tight Analysis of Extra-gradient and Optimistic Gradient Methods For Nonconvex Minimax Problems"
_NeurIPS.cc/2022/Conference — NeurIPS 2022 Accept_

### Official Review · Reviewer_C83W · 2022-07-11

**Rating:** 7
**Confidence:** 4
**Soundness:** 4 excellent
**Presentation:** 4 excellent
**Contribution:** 3 good

**Summary:**

This paper shows the convergence of Extragradients methods in the context of nonconvex-(strongly)concave minimax optimization.


**Questions:**

- In your experiments: Can you report the performance of each methods (in terms of Wasserstein distance between the generated and the data distribution)?
- It seems to me that reporting the actual norm of the gradient (i.e the sum of the norm of the gradients of the discriminator and the generator) would be a more relevant metric of convergence than both norms independently.


**Limitations:**

The authors mention the slight gap between the lower and the upper bound for NS-SC problems and the missing lower bound for OGDA

**Strengths And Weaknesses:**

Strenghts:
- The authors propose tight upper and lower bounds for their analysis
- They study the Extragradient and the Optimistic method.
- They propose some non-trivial experiments (e.g. GANs on a toyish dataset)  to compare different optimization methods. The experiment is non-trivial because the architectures considered are non-convex.

weaknesses:
- The lower bounds shown in the paper only regard the methods considered and not a class of methods (e.g. first-order methods or p-SCLI methods)
- Minor issues: There are some formatting issues (page 2). The page number is overlapping the text

---

> ### Author Response · Authors · 2022-08-02
> **Response to Reviewer C83W**
>
> Thank you for the positive feedback. Your questions about the experiments are kindly addressed below.
>
> ```
> Q.1 In your experiments: Can you report the performance of each method (in terms of Wasserstein distance between the generated and the data distribution)?
> ```
>
> **A.1:** Theoretical guarantees in our paper are based on the convergence to stationary point measured by the gradient norm, which is why we focused on evaluating the gradient norm in our experiment. However, we totally agree that evaluating the performance of different optimization methods in terms of Wasserstein distance between the generated and real distribution provides good insight into the applicability of these methods in practice. Therefore, we will add the Wasserstein metric evaluation following the setup in [R1] (Figure 4 in their paper) and will include it in the revised version. We appreciate the suggestion.
>
> ```
> Q.2 It seems to me that reporting the actual norm of the gradient (i.e. the sum of the norm of the gradients of the discriminator and the generator) would be a more relevant metric of convergence than both norms independently.
> ```
>
> **A.2:** Thanks for mentioning this point. We agree that the actual gradient norm is a more relevant measure and will also add this metric to the experiment. Per your comment, we ran experiments by measuring the actual norm, and the plot can be found in this [link](https://anonymous.4open.science/r/figures-EE4D/total_gradient_norm.png), including an evaluation of different algorithms in terms of the mentioned metric. As it can be seen, we observed that OGDA still outperforms all other methods, showing its fast convergence empirically.
>
>
> [R1] Li, Haochuan, Farzan Farnia, Subhro Das, and Ali Jadbabaie. "On Convergence of Gradient Descent Ascent: A Tight Local Analysis." ICML, 2022.

---

> > ### Comment · Reviewer_C83W · 2022-08-03
> > **Thank you for your answer**
> >
> > Thank you for your answer. I would like to say I am slightly disappointed that the author did not include a revision of the paper with the experiments mentioned in their response.
> > However, I maintain my score.

---

> > > ### Author Response · Authors · 2022-08-03
> > > **Revision of the paper**
> > >
> > > Many thanks for your feedback on the rebuttal. We totally agree it would have been much better to upload a revised version of the paper with changes, but we noticed in the very last moment that we can upload a revised version. We commit to incorporate all the changes we mentioned in the responses and apologize for any inconvenience this might have caused.

---

### Official Review · Reviewer_8N1p · 2022-07-11

**Rating:** 6
**Confidence:** 4
**Soundness:** 3 good
**Presentation:** 4 excellent
**Contribution:** 3 good

**Summary:**

The paper, proposes a convergence analysis of Optimistic gradient descent ascent (OGDA) and Extragradient (EG) methods for two classes of min-max optimization problems the nonconvex-strongly-concave (NC-SC) and nonconvex-concave (NC-C). The authors also provide lower bounds showing the tightness of their convergence guarantees.

**Questions:**

Below i include some questions related to the weaknesses of the paper:

1. if I did not miss its definition, i believe $\bar{x}$ is only defined in Algorithm 1 (GDA). However the authors provide $\bar{x}$ as the output of OGDA and EG as well. Can the authors elaborate more on this? Why one need to have $\bar{x}$ as the output of OGDA and EG and not just simply use the last iterate of these methods (note that in the Algorithms 2 and 3 no output is given)?

2. In section 4.2 what is the point of mentioning the following sentence?
"From the above assumption we note when f is merely concave in y, we have to assume the domain for dual variable is bounded, and hence it requires projection in the updating rule of y:" The two algorithmic updates are exactly the same with the update rules first introduced in section 3.1, right?

3. In the current problem (1), it is assumed that there is unbounded x, and y \in Y. How the paper's theoretical results would be affected in the scenario that y \in R^d ( no constraints) or in the scenario that x \in X where X is bounded convex set (classical constrained min-max optimization problem). In other words, is there any particular reason (in terms of proof techniques) that the specific setting is considered?

4. On Experiments:
In the experiments you use the setting from [29] but you did not compare against the algorithms from [29]. Was there any particular reason for this? In addition i believe that parameter alpha and beta were never properly defined in the main paper but they are referenced extensively in the experiments related to generalized OGDA?

**Limitations:**

Yes limitations have been addressed.

**Strengths And Weaknesses:**


Strengths:

The paper is well written and the main contributions of the work are well presented. I went through the main proofs and the important steps looks also correct. I really like the idea of including Proof sketches, before the actual proofs in the appendix. This add pedagogical value to the presentation and makes it easier to follow.

In terms of presentation I also like Table 1 with the summary of main theoretical contributions and comparison of the proposed results with previous papers. In this table  would suggest to include also a line with the lower bounds. This would highlight better the tightness of your results as one will be able to see early in the paper the two main contributions of the work (convergence rates and lower bounds), in the a single table.

Weaknesses:

In general i like this paper and i believe its contributions could be valuable to the ML community. To the best of my knowledge this is the first paper that analyze OGDA and EG for NC-SC and NC-C min-max problems. However i notice some issues/weaknesses that will require further clarification.

There are inconsistencies between the main paper and the appendix, in terms of notation and presentation of the results. These inconsistencies raised some questions the robustness of the theoretical results. Let me add a few details below:

1. In the statement of the theorems in the main paper the quantity $\Phi(x_1) -min_x \Phi(x)$ is defined as $\Delta_\Phi$ while in the appendix the same quantity is $\delta$. I would suggest the authors do a thorough pass and fix these issues. Statements of Theorems and their proofs need to use the same notation.

2. The main theoretical results in the main theorems are presented in terms of gradient complexity, however the proofs of main Theorems  stop in equations (49) and (75). The authors should explain (even if this might be trivial) in more detail and provide the exact derivation of how one can obtain the main Theorem (the gradient complexity) from the end of their proofs.

3. As we can see for the proof to work the algorithms should be run with specific step-size selection. In the current statement of the Theorems the exact values of the step-size for OGDA and EG are hidden in the big $\Theta$ expression. The authors should add the exact step-size values that used in the proofs (at the moment the Theorems are more abstract than what should be).

4. The step-sizes of the main theorems in the section 4.2 (Nonconvex-concave minimax problems) are not part of the statement of the theorems. This should be added in the updated version.

5. More clarifications are needed for some parts. I include a few questions about these points below.

6. Minor comments:

(a)  Definition 3.3. and stochastic definition of in line 120 use both "g". This might be confusing for some readers.
(b) Theorem 4.5 is about GDA but in the statement the author refer to Theorem 4.2which is a bit confusing.
(c) The title could be updated to include "nonconvex-concave" instead of simple nonconvex as it is not. At the moment the reader expects that the methods proposed solve nonconvex nonconave problem (potentially with extra structured).

6.There are several recent closely related missing references (could be added in corresponding paragraphs of Section 2):

[1] Abernethy, J., Lai, K. A., Wibisono, A.. Last-iterate convergence rates for min-max optimization: Convergence of hamiltonian gradient descent and consensus optimization. In Algorithmic Learning Theory (pp. 3-47). PMLR

[2] Loizou, N., Berard, H., Jolicoeur-Martineau, A., Vincent, P., Lacoste-Julien, S., Mitliagkas, I. . Stochastic hamiltonian gradient methods for smooth games. In International Conference on Machine Learning (pp. 6370-6381). PMLR.

[3] Gorbunov, E., Berard, H., Gidel, G., Loizou, N. . Stochastic extragradient: General analysis and improved rates. In International Conference on Artificial Intelligence and Statistics (pp. 7865-7901). PMLR.

[4] Gorbunov, E., Loizou, N., Gidel, G.. Extragradient method: O (1/K) last-iterate convergence for monotone variational inequalities and connections with cocoercivity. In International Conference on Artificial Intelligence and Statistics (pp. 366-402). PMLR.

[5] Hsieh, Y. G., Iutzeler, F., Malick, J., Mertikopoulos, P. . Explore aggressively, update conservatively: Stochastic extragradient methods with variable stepsize scaling. Advances in Neural Information Processing Systems, 33, 16223-16234.

[6] Cai, Y., Oikonomou, A.,  Zheng, W.. Tight Last-Iterate Convergence of the Extragradient and the Optimistic Gradient Descent-Ascent Algorithm for Constrained Monotone Variational Inequalities. arXiv preprint arXiv:2204.09228.

[7] Loizou, N., Berard, H., Gidel, G., Mitliagkas, I.,  Lacoste-Julien, S.. Stochastic gradient descent-ascent and consensus optimization for smooth games: Convergence analysis under expected co-coercivity. Advances in Neural Information Processing Systems, 34, 19095-19108.

[8] Beznosikov, A., Gorbunov, E., Berard, H.,  Loizou, N. . Stochastic gradient descent-ascent: Unified theory and new efficient methods. arXiv preprint arXiv:2202.07262.

---

> ### Author Response · Authors · 2022-08-02
> **Response to Reviewer 8N1p**
>
> We appreciate your constructive comments. We acknowledge the inconsistency between the main body and the appendix. We will apply your comments to make the appendix more consistent. The questions you have raised are addressed below and will be incorporated in the revised version:
>
> ```
> Q1: More explanation on having randomly selected iterate $\bar{x}$ as the output of OGDA/EG.
> ```
>
> A1: Our convergence rates are obtained by finding an upper bound on the average of iterates' primal function (or Moreau envelope function) gradient norm. In other word, the convergence can be guaranteed for randomly selected iterate ($\bar{x}$) or the best iterate. Choosing $\bar{x}$ as the final output of the algorithm is actually standard in the literature of NC-C/NC-SC minimax, and we followed the same measure in our paper. To the best of our knowledge, last-iterate or exact-iterate convergence rate are missing for NC-C/NC-SC minimax optimization methods including EG/OGDA/GDA and that would be an interesting future direction.
>
> ```
> Q2: In Section 4.2 what is the point of mentioning the following sentence? "From the above assumption we note when f is merely concave in y, we have to assume the domain for dual variable is bounded, and hence it requires projection in the updating rule of y:" The two algorithmic updates are exactly the same with the update rules first introduced in section 3.1, right?
> ```
>
> A2:  Yes, the two algorithmic updates are exactly the same. By the sentence you mentioned, we actually meant that when the objective $f$ is merely concave, we need the dual variable domain to be bounded since otherwise the Moreau envelope function will not be well-defined (This is shown in Lemma 3.6 in [23]). On the other hand, when $f$ is strongly-concave with respect to dual variable, our proof for convergence bound holds even when the dual variable domain is unbounded, e.g. $\mathcal{Y} = \mathbb{R}^d$. Per your comment, we will make the statement more clear in the revised version. Thanks!
>
> ```
> Q3: In the current problem (1), it is assumed that there is unbounded $x$, and $y \in \mathcal{Y}$. How the paper's theoretical results would be affected in the scenario that $y \in \mathbb{R}^d$ ( no constraints) or in the scenario that $x \in \mathcal{X}$ where $\mathcal{X}$ is bounded convex set (classical constrained min-max optimization problem). In other words, is there any particular reason (in terms of proof techniques) that the specific setting is considered?
> ```
>
> A3: For NC-SC setting, there is no restriction on dual variable domain ($\mathcal{Y}$), and our proofs work when $\mathcal{Y}$ is either $\mathbb{R}^d$ or a bounded convex set. For NC-C case, however, as partially explained in Q.2, if $\mathcal{Y}$ is not a bounded convex set, the Moreau envelope would not be well defined, thus our analysis can not be applied beyond this setting. Gradient norm of Moreau envelope function has been established as one of the key optimality criterion for NC-C, and this is the reason we chose to work with this measure, and considering bounded convex set for dual variable domain in NC-C. For primal variable $\mathbf{x}$, the most of existing literature in NC-C and NC-SC assumed $\mathcal{X} = \mathbb{R}^d$, and we followed the same setting. Some practical NC-SC/NC-C minimax problems where primal variable is unconstrained, and dual variable domain is bounded convex set can be found in Section~$5$ of [34] including fair classifier, and robust neural network training. However, we believe that main steps of the proofs will stay the same assuming the primal variable is restricted to a bounded convex  set.
>
> ```
> Q4: On Experiments: In the experiments you use the setting from [29] but you did not compare against the algorithms from [29]. Was there any particular reason for this? In addition i believe that parameter alpha and beta were never properly defined in the main paper but they are referenced extensively in the experiments related to generalized OGDA?
> ```
>
> A4: Our experiment is based on the [44], not [29]. Stochastic smoothed AGDA [44] is introduced as the fastest single loop algorithm for NC-SC theoretically, and this was the reason we chose [44] to compare this algorithm with OGDA in practice. Moreover, we already compared our algorithm with all algorithms in [44] as it is shown in Figure 1(a). We acknowledge that experiment in [44] is actually based on [28], however the problem setup and algorithms in [28] are totally different than ours as they considered variance reduction, and Hamilton gradient descent, while our problem setup is clearly different. Also, note that even [44] does not compare with the algorithms in [28].
>
> Moreover,  the parameters $\alpha$ and $\beta$ are supposed to represent the ratio between the correction term, and the current gradient for generalized OGDA algorithm. We acknowledge these parameters  should be introduced earlier in paper, and we will ensure that to make it well defined in the updated version.

---

### Official Review · Reviewer_yYTE · 2022-07-12

**Rating:** 6
**Confidence:** 4
**Soundness:** 3 good
**Presentation:** 3 good
**Contribution:** 3 good

**Summary:**

This paper provides theoretical analysis of optimistic gradient descent ascent (OGDA) and extragradient (EG) on the convergence and its tightness. For both OGDA and EG, two families of problems are considered, i.e., nonconvex-strongly-concave (NC-SC) and nonconvex-concave (NC-C), and two settings are analyzed, i.e., deterministic and stochastic (minibatch), respectively, leading to totally four specific conditions.

The convergence analysis of both algorithms looks under a unified framework. For example, Lemma A.4 and Lemma A.8 create similar telescoping summation to start analysis of OGDA and EG under NC-SC, respectively. Similar situations can be seen in Lemma B.2, Lemma B.8, Lemma B.15 and Lemma B.20, where Moreau Envelope is first used to start for deterministic OGDA, stochastic OGDA, deterministic EG and Stochastic EG, respectively.

Compared to previous convergence rate of GDA in the same four conditions, this paper achieve the same results of OGDA/EG in terms of gradient complexity (Table 1). In addition, tightness of these convergence results is analyzed by comparing with the lower bounds deriving from the designed hard function instances, showing that convergence results are tight.


**Questions:**

If I did not miss anything, there may be something inconsistent (and minor if it is) when using Moreau Envelope for NC-C analysis. In Lemma B.2, after the final inequality, the proof says that one can use $\|| \hat x_{t-1} - x_{t-1} \|^2 = 1 / (2\ell) \||\nabla \Phi_{1/(2\ell)} (x_{t-1}) \||^2$. Given the definition of $\hat x_{t-1}$ in the beginning of Proof for Lemma B.2, I think one has $\hat x_{t-1} – x_{t-1}=(1/2\ell) (\nabla \Phi_{1/(2\ell)} (x_{t-1}))$ (this result can also be found in [7], the equation right above Figure 1 therein), so $\|| \hat x_{t-1} – x_{t-1} \||^2=(1/4\ell^2) \|| \nabla \Phi_{1/(2\ell)}(x_{t-1}) \||^2$, or $\|| \hat x_{t-1} – x_{t-1} \||=(1/2\ell) \|| \nabla \Phi_{1/(2\ell)}(x_{t-1}) \||$. In fact, the latter equation is shown in Proof of Lemma B.20, Line 982.

Similar concern occurs in Lemma B.8, Lemma B.15 and Lemma B.20. If the above dependency of $\ell$ changes, then it may change a little bit on the lower bound for $\eta_x$ in Theorem B.6 (NC-C, deterministic OGDA), Theorem B.12 (NC-C, stochastic OGDA), Theorem B.18 (NC-C, deterministic EG) and Theorem B.23 (NC-C, stochastic EG).




**Limitations:**

I think the authors adequately addressed the limitations and potential negative societal impact of their work

**Strengths And Weaknesses:**

Originality:
This paper provides convergence analysis of OGDA and EG for nonconvex problems, i.e., NC-SC and NC-C, matching that of GDA in previous studies, which offers new understanding to these algorithms. Tightness analysis also gives further understanding to convergence results.

Specifically, to deal with the gap between OGDA/EG and existing analysis such as [24] of GDA for NC-SC and NC-C, as well as [33] of OGDA/EG for SC-SC, this paper constructs several new bounds to derive the convergence result. For example, since the update is different from GDA, the bound of $\||y_i-y^*(x_i)\| |^2$ developed in [24] is not available. Instead, this paper crafts new upper bound for term $r_t$, which involves $\||z_{t+1}-y^*(x_t)\| |^2$ and $\||y_t-y_{t-1}\||^2$. These ideas are particularly discussed in Section 4.3 conceptually. Proof part is complete and self-contained.

Quality:
This paper is technically sound. Proofs look almost (please see ‘Questions’ part for my concern) correct to me.


Clarity:
This paper is clear and easy to follow. However, it looks a little bit long (65 pages totally) and some results may be further merged/organized, so that it could be somehow explicitly show the unified theoretical framework of both algorithms.


Significance:
This paper provides significant insights to OGDA and EG algorithms for nonconvex (strongly) concave min-max problems.

---

> ### Author Response · Authors · 2022-08-02
> **Response to Reviewer yYTE**
>
> We appreciate your careful review. Your question regarding the proof of NC-C analysis is kindly addressed below, and thank you for mentioning this point.
>
> ```
> Q1:  Concerns about using a Moreau Envelope property for NC-C analysis.
> ```
>
> A1: In proof of Lemma B.2, we actually used the fact that $ || \hat x_{t-1} - x_{t-1} || = \frac{1}{2 \ell} || \nabla \Phi_{\frac{1}{2 \ell}} (x_{t-1})   ||$, and the last equation (Line 824), $ || \hat x_{t-1} - x_{t-1} ||^2 = \frac{1}{2 \ell} || \nabla \Phi_{\frac{1}{2 \ell}} (x_{t-1}) ||^2 $, is a typo, and should be replaced with $ || \hat x_{t-1} - x_{t-1} || = \frac{1}{2 \ell} || \nabla \Phi_{\frac{1}{2 \ell}} (x_{t-1})   ||$. Also, in RHS of the last inequality above Line 824, there is another typo in which the term $- \frac{\eta_x \ell}{2} ||\hat x_{t-1} - x_{t-1} ||^2 $ should be replaced with $ - \frac{\eta_x \ell^2}{2} || \hat x_{t-1} - x_{t-1} ||^2 $. It can be easily checked that Lemma B.2 still holds after fixing the two mentioned typos in the proof. A similar argument is valid for Lemmas B.8, B.15, and B.20. Therefore, we believe by fixing the mentioned typos, our proof will be complete without changing the lemmas and theorems.  We appreciate your careful reading, and apologize for typos and will fix them in the revised version.

---

### Official Review · Reviewer_8dwY · 2022-07-16

**Rating:** 6
**Confidence:** 4
**Soundness:** 3 good
**Presentation:** 3 good
**Contribution:** 3 good

**Summary:**

This paper analyzes the convergence rates of the two-time-scale versions of extra-gradient (EG) and optimistic gradient (OGDA) methods
for nonconvex-strongly-concave (NC-SC) and nonconvex-concave (NC-C) settings. Similar rate analysis has been already studied for a two-time-scale variant of GDA in [Lin et al., 2020], and this paper's rates of two-time-scale EG and OGDA methods are found to be (almost) equivalent to those of GDA. This equivalence is new and interesting, but might look disappointing in some sense, especially considering that OGDA performs well in practice as illustrated in Section 6. The authors further provide lower bounds of GDA/OGDA/EG, and confirm that [Lin et al., 2020] and this paper's rates are tight.

A simple WGAN experiment illustrate that OGDA might outperform existing algorithms in some practical NC-(S)C problems. Further empirical investigation on different stepsizes for the current gradient and correction term of OGDA suggests that certain different stepsizes might empirically further accelerate OGDA.

**Questions:**

- Table 1: ALSET [2] and [Nouiehed et al., 34] are basically similar, so I think [34] should also be included, especially considering that [34] also has a result on the NC-C setting. Let me know if I am missing something.
- Line 39: Among [32,9,33], [9] is the one that discusses the stochastic setting, and it seems to only consider the SC-SC case. I suggest adding another reference on the stochastic C-C case, if there is any.
- Line 49: The rates in the paper are in terms of the randomly chosen point $\bar{x}$ (or the best iterate for the NC-C setting), but here it is called the ergodic average.
- Contributions: I believe that the contributions part can be parsed so that each contribution is more clearly given to the readers.
- Line 76: the the
- Line 86: preforming
- Line 97: The nonconvex-"non"concave setting does not seem to be discussed here.
- Line 188-192: Could you explain this in more details? I was also not able to find its related part in [23].
- Line 235: I am not following why OGDA/EG has an inherent nice descent property on concave function, which GDA does not have. Could you write down such property explicitly? (If it is in the appendix, let me know where it is.) Why don't we see such difference on the descent property (and its resulting rates) in NC-SC setting? In my viewpoint, the $\hat{\Delta}_0$ terms are not the "significant" ones, compared to the other.
- Line 277: leaning rate (this also appears in other places)
- Figure 1: What is the step sizes for OGDA? Do they match the theoretically chosen step size in theorems?
- Line 340: generlaztion

**Limitations:**

- In conclusion, the authors state that the general lower bound for OGDA in NC-C setting and stochastic setting is missing. Also, the authors point out that there is a gap between the lower and upper bounds for NC-SC problems. I think the authors reasonably addressed this work's limitations, including the rate equivalence between GDA and OGDA/EG.

**Strengths And Weaknesses:**

- S1: It is the first rate analysis of EG and OGDA for NC-SC and NC-C settings.
- S2: Both stepsize-dependent and stepsize-independent lower bounds of GDA/EG/OGDA are new, and provide how much tight the derived upper bounds are.

- W1: Theorems state that OGDA/EG converge to an $\epsilon$-stationary point, but they only guarantee finding it. This is not a weakness, and it just needs correction. A minor weakness of this paper's analysis of OGDA/EG that I want to point out is that they do not guarantee finding an "exact" solution as we further iterate, similar to GDA [23].
- W2: Theoretically, this work only suggests that OGDA/EG is as much good as GDA.

---

> ### Author Response · Authors · 2022-08-02
> **Response to Reviewer 8dwY**
>
> We appreciate your comments and are sorry for any inconvenience caused by the grammatical and writing errors. We will address such mistakes and improve the writing quality for the final version. Moreover, we will gladly address your questions and incorporate them in the revised version as detailed below:
> ```
> Q1: (Line 188-192) Comparison between gradient oracle complexity of two-time scale SGDA versus stochastic OGDA/EG for NC-SC setting.
> ```
>
> A1: Thanks for raising this point. In Equation 49 (appendix), we derived the final upper bound of stochastic OGDA in terms of $M_x$ and $M_y$, the batch size for primal and dual variables in computing stochastic gradients, respectively. We can conclude from Equation 49 that for reaching an $\epsilon$-stationary point, $M_x = O (\frac{1}{\epsilon^2}) $, and $M_y = O (\frac{ \kappa}{\epsilon^2})$. However, for the analysis of  SGDA in Theorem 4.5 of [23], the batch size for both primal and dual variables is the same and equal to $O(\frac{\kappa}{\epsilon^2})$. Since the number of iterations has shown to be $T = O(\frac{\kappa^2}{\epsilon^2}) $ in both OGDA/EG (Theorem 4.4 in our paper)  and GDA (Theorem 4.5 in [23]), our analysis for stochastic OGDA shows an improvement in terms of primal gradient complexity. However, we agree that this paragraph needs more clarification as Theorem 4.4 in our paper is written in the case of $M_x = M_y$ and does not show the improvement we achieved in Equation 49 (appendix). We will modify this paragraph and the statement of Theorem 4.4 and add the clarification above in the subsequent version of this paper.
>
> ```
> Q2: (Line 235)  More explanation of inherent nice descent property on concave function.
> ```
>
> A2: Take OGDA for example. The key step in NC-C analysis is to bound $\Phi(x_t) - f(x_{t},y_{t})$. In OGDA proof, we split this into  bounding the following:
> $  \Phi(x_{t }) - f(x_{t }, y_{t }) \leq f(x_{t }, y^*(x_{t }))-f(x_{s}, {y}^*(x_{t }))+f(x_{s},{y}^*(x_{s})) - f(x_{t },{y}^*(x_{s}))  + f(x_{t },{y}^*(x_{s})) - f(x_{t },{y}_{t }).$
>
> For the last term $f(x_{t },y^*(x_{s})) - f(x_{t},y_{t })$, OGDA can guarantee its convergence without bounded gradient assumption on $y$. However, for GDA, it requires bounded gradient assumption on $y$ to show the convergence of this term, and without such assumption, we can only show the convergence of $f(x_t,y^*(x_{s})) - f(x_{t },y_{t+1})$, so Lin et al split the  $\Phi(x_{t }) - f(x_{t },y_{t})$ as follow:
> $\Phi( x_t) - f( x_{t }, y_{t }) \leq f(x_{t }, y^*( x_{t }))-f( x_{t}, y^*( x_{s}))+f(x_{t+1}, y_{t+1}) - f( x_{t }, y_t)  + f( x_{t }, y_{t+1}) - f( x_{t+1}, y_{t+1})+ f( x_t, y^*(x_s)) - f(x_t, y_{t+1}) $.
> Hence they reduce the problem to bounding $f(x_{t },y^*(x_{s})) - f(x_{t },y_{t+1})$. Unfortunately, they have to pay the price for the extra term $f(x_{t+1},y_{t+1}) - f(x_t ,y_t)$. We will add a clarification to the revised version to make this clear. Thanks for raising this question.
>
> ```
> Q3: What is the step sizes for OGDA? Do they match the theoretically chosen step size in theorems?
> ```
>
> A3: Regarding the step sizes in the experiment, we observed that using $\eta_x =  \eta_y = 0.05$ leads to the best convergence in terms of the gradient norm. We note that this choice of learning rates is not exactly what our theorems suggest  (i.e, using a large value for $\eta_x$ when the condition number is large), because the objective in our experiments does not fulfill the assumptions in convergence analysis due to presence of ReLU as primal objective. We will conduct experiments with a smooth activation to evaluate the effect of  different learning rates as dictated by our theorems when the condition number is large in revised version.
>
> ```
> Q4: Concerns regarding the Related Work section.
> ```
>
> A4: We will update Table 1 with the suggested reference, and thank you for bringing these papers to our attention.
>
> Regarding the nonconvex-(non)concave subsection in the related work section, we meant to name this section nonconvex-concave, and the title for this subsection is a typo. The nonconvex-nonconcave result for EG/OGDA is discussed in the first subsection of the related work. However, we believe it would be helpful to add a separate nonconvex-nonconcave subsection to further complete the related work, which we will gladly include.

---

> > ### Comment · Reviewer_8dwY · 2022-08-09
> > **Thanks for the response**
> >
> > I decided to maintain my score, but like the reviewer C83W, I also think the authors could have done better by providing some additional preliminary experimental results (at least in text). Regarding Q2, I now better understand what authors were trying to say here, and I think it was impossible to deduce such explanation from the text "an inherent descent property". This part is especially important as it distinguishes between GDA and OGDA, so I hope this becomes crystal clear in the revision.

---

### Meta-Review · Area_Chair_Jzqz · 2022-08-23

**Recommendation:** Accept
**Confidence:** Certain

**Metareview:**

The reviewers appreciate the novel theoretical contribution to the convergence analysis of OGDA and EG methods in the settings of the nonconvex-strongly-concave (NC-SC) and nonconvex-concave (NC-C) as well as the lower bounds showing the tightness of their convergence guarantees. I recommended its acceptance accordingly. The suggestions of the reviewers should be included in the revised version.

**Award:**

No

---

### Decision · Program_Chairs · 2022-09-14

Accept